# Callus Induction and Adventitious Root Regeneration of Cotyledon Explants in Peach Trees

Lingling Gao [1,2,3], Jingjing Liu [1,3], Liao Liao [1,2], Anqi Gao [1,3], Beatrice Nyambura Njuguna [1,3], Caiping Zhao [4], Beibei Zheng [1,2,*] and Yuepeng Han [1,2,*]

1 CAS Key Laboratory of Plant Germplasm Enhancement and Specialty Agriculture, Wuhan Botanical Garden, The Innovative Academy of Seed Design of Chinese Academy of Sciences, Wuhan 430074, China
2 Hubei Hongshan Laboratory, Wuhan 430070, China
3 University of Chinese Academy of Sciences, 19A Yuquanlu, Beijing 100049, China
4 College of Horticulture, Northwest Agriculture and Forestry University, Yangling 712100, China; zhcc@nwsuaf.edu.cn
* Correspondence: zhengbeibei@wbgcas.cn (B.Z.); yphan@wbgcas.cn (Y.H.)

**Abstract:** Callus induction is a key step in establishing plant regeneration and genetic transformation. In this study, we present a comprehensive large-scale investigation of the callus induction rate (CIR) in peach trees, which revealed significant variability within the peach germplasm. Notably, the late-maturing cultivars exhibited significantly higher levels of CIR. Moreover, cultivars characterized as having high CIR exhibited potential for the development of adventitious roots (ARs) during callus induction, and a positive correlation was observed between CIR and the ability to regenerate ARs. However, long-term subculture callus lost root regeneration capacity due to changes in cellular morphology and starch and flavonoid content. Additionally, *PpLBD1* was identified as a good candidate gene involved in the regulation of callus adventitious rooting in peach trees. Our results provide an insight into the mechanisms underlying callus induction and adventitious root development and will be helpful for developing regeneration systems in peach trees.

**Keywords:** *Prunus persica*; cotyledon; adventitious root regeneration; long-term subculture



## 1. Introduction

Peaches (*Prunus persica* L.) are widely cultivated in temperate regions as an important economic fruit crop. The trees' juvenility period is relatively short and the species has a small haploid genome size, approximately 230 Mb [1,2]. However, it is well known that the genetic transformation of the species is very difficult [3]. To date, a stable genetic transformation system is not available for peaches, which significantly hinders its application as a genetic research model for Rosaceae fruit trees [4].

Callus induction and subsequent plantlet regeneration are the essential prerequisite for genetic transformation [5]. Callus induction depends on cell fate transition that turns the somatic state into pluripotency [6]. Callus can be derived from many kinds of explants such as leaves, mature embryos, stems, and immature embryos, but different types of calli have a significant difference in embryogenic potential [7]. In peach trees, callus induced from immature embryos is able to develop shoots, but mature embryos show callus formation without shoot development [8]. However, in apple trees, embryogenic callus can be efficiently induced from leaf explants from the cultivar Royal Gala, which facilitates the establishment of a transformation system [9]. As callus induction materials, cotyledon explants have been proved to have high frequency regeneration due to their higher content of meristematic tissue, lower level of contamination, and improved browning [10–13]. Overall, leaf and cotyledon explants are the main sources of callus induction and subsequent plantlet regeneration [14,15]

In peach trees, callus induction from leaf and cotyledon explants has been widely reported, but there are few reports of successful plantlet regeneration from callus culture [16–18]. Recently, Xu et al. [19] developed a fast and efficient root transgenic system through *Agrobacterium tumefaciens*-mediated transformation in peach trees. However, transgenic roots induced by *Agrobacterium* are difficult to develop lateral roots (LRs) as shown in our study [19]. Thus, root differentiation seems to be a barrier for plant regeneration in peach trees. Moreover, adventitious root emergence is a crucial step in stem cutting propagation of fruit tree rootstocks. However, rooting difficulty of stem cuttings has been reported to be a serious problem in asexual propagation of peach rootstocks [20]. There are few reports to date on the adventitious root development and growth mechanism of peach trees [16,21].

Adventitious roots are essential for plant growth due to their functions of expanding plant absorption area and supporting plants [22]. The mechanism of adventitious root development in model plants such as *Arabidopsis thaliana* (L.) Heynh. and *Oryza sativa* L. have been extensively studied. For instance, in *A. thaliana*, members of the lateral organ boundary domain *(LBD)* gene family play an important role in plant organ development [23]. Auxin response factors (ARFs) such as *AtARF7* and *AtARF19* are involved in adventitious root occurrence by triggering the transcription of *AtLBD16* and *AtLBD18* [24]. However, auxin/indole-3-acetic acid protein (AUX/IAA) interacts with ARFs to inhibit ARF-induced AR formation [25]. In rice, the LBD gene crown rootless1 (*CRL1*) is required for adventitious root origination [26,27]. In addition to LBD and ARF transcription factors (TFs), the basic helix–loop–helix (bHLH) TF family was also reported to regulate root formation [28,29].

Although numerous peach somatic tissues, for instance, leaves, stems and calyxes, have been successfully applied to induce callus [30–32], the genetic mechanisms underlying callus induction have not yet been illustrated. In this study, we evaluated diversity in the callus induction rate among peach cultivars. Interestingly, adventitious roots were often developed during callus culture, and there was a positive correlation between callus induction rate and root regeneration capacity. Moreover, we identified candidate genes for callus-based adventitious root formation. Our results not only provide insight into the mechanism of adventitious root formation, but will also contribute to the establishment of a genetic transformation model in peach trees.

## 2. Material and Methods

### 2.1. Plant Material and Growth Conditions

The peach cultivar ZYT used in this study was grown at Hubei Academy of Agricultural Sciences, Wuhan, Hubei Province, China, while the other 99 peach cultivars, including 2 wild relatives, 8 landrace accessions, and 89 modern cultivars (Table S1), were grown at Northwest A&F University, Yangling, Shanxi Province. The surface sterilization method used was optimized based on a previous study [19]. Firstly, the pits of ripe fruits were collected by peeling away the flesh with a knife and subsequently immersed in a solution of sodium hypochlorite and water ($v/v$ = 1:3) for 0.5 h. Later, the pits were thoroughly rinsed three times with distilled water, left to air-dry at room temperature, and then stored in the refrigerator at 4 °C. Two months later, the endocarp was removed and the seeds were sterilized with 70% ethanol for 2 min. The seeds were transferred to a sterile triangular flask and submerged in a solution of sodium hypochlorite and water ($v/v$ = 1:3) for a duration of 0.5 h, all while placed on an ultraclean platform. After 10 rinses with sterile water, the seeds were incubated in sterile water at 25 °C for 15 h.

The coat of the seeds was removed with a scalpel, and the embryos and hypocotyls were discarded. The cotyledons were cut into rectangular pieces of 1 cm × 0.5 cm. The pieces were then placed in callus induction medium (CIM) as previously reported [30]. The pH of the medium was adjusted to 5.8 and sterilized using an autoclave at 121 °C for 20 min. Three biological replicates of each cultivar were created, and each replicate contained at least two cotyledons.

## 2.2. Estimation of Callus Induction Rate and Rooting Rate in Peach trees

After 3 weeks of culture in CIM, the cotyledon was photographed using a Nikon SMZ25 microscope. The micrographs were used to estimate the size of the cotyledon and its callus with ImageJ v1.8.0. CIR was calculated using the following formula: $CIR = \frac{callus\ area\ \times\ 100\%}{original\ cotyledon\ area}$. After being cultured in CIM for 6 weeks, the number of callus-induced roots was recorded. Five-year-old subcultured ZYT callus of was used as a control.

## 2.3. Histological and Ultrastructural Analyses

Callus induced from cotyledons at different stages was examined via scanning electron microscope (SEM). After fixation with 2.5% glutaraldehyde, the callus was freeze-dried. A Leica EMACE ion-sputtering coater (Leica, Wetzlar, Germany) was used to coat a layer of gold powder onto the surface of the callus. Then, a desktop scanning electron microscope (Hitachi, TM3030, Tokyo, Japan) was used to scan the surface structure of the callus.

For analysis via transmission electron microscopy (TEM), the callus was fixed with 2.5% glutaraldehyde and then submerged in ethanol with differing gradient concentrations for 3 h. The fixed callus was cut into slices and stained, first with aqueous uranyl acetate and then with lead citrate. The stained section was scanned using transmission electron microscope (JEOL, Inc., Boston, MA, USA).

## 2.4. Measurement of Plant Metabolites

Fresh callus (100 mg) was ground into powder in liquid nitrogen, then the target product was extracted using a mixture of methanol:water ($v/v$ = 80:20) at 4 °C. Flavonoid content and total phenol content were measured using the Plant Flavonoids Assay Kit (BC1335, Solarbio, Beijing, China) and the Total Phenol Assay Kit (BC1345, Solarbio, Beijing, China), respectively. The measurement of soluble sugars was based on our previous report [33].

## 2.5. RNA Extraction, Quantitative Real-Time PCR and Transcriptome Analysis

Callus samples of approximately 100 mg were collected and ground in liquid nitrogen for RNA extraction. RNA Rapid Extraction Kit (Megan, Guangzhou, China) was used for total RNA isolation. A Primer Script™ RT reagent Kit with gDNA Eraser (TransGen, Beijing, China) was used for first-strand cDNA synthesis. Quantitative real-time RT-PCR (qRT-PCR) was performed using SYBR premix Ex Taq II Kit (YEASEN, Shanghai, China) according to the manufacturer's instructions to yield a final volume of 20 μL. qRT-PCR was checked via StepOne Plus (ABI), and the internal reference gene was *PpTEF2* [34]. The amplification procedure was 95 °C for 30 s, 95 °C 5 s, and 60 °C 5 s, over 40 cycles. Relative gene expression level was measured according to the cycle threshold (Ct) $2^{-\Delta\Delta Ct}$ method [35]. Three replications were conducted for each treatment. Table S7 lists the primers used in the experiment.

## 2.6. Transcriptome Analysis and Identification of Differentially Expressed Genes (DEGs)

RNA samples for library construction were prepared following the abovementioned method, with each sample consisting of three biological replicates. RNA sequencing was performed using the Illumina HiSeq2500 platform. Clean reads were obtained by removing adapter sequences and low-quality reads. The HISAT2 v2.1.0 software program was utilized to align the clean reads to the reference genome [2]. The DESeq2 v1.22.2 software program was used to analyze the differential expression of each group [36,37]. Fragments per kilobase of transcript per million fragments mapped (FPKM) was utilized as an index to quantify the level of gene expression. Genes with $|log2fold\ change| \geq 1$ and a false discovery rate (FDR) < 0.05 were considered DEGs, and FDR was estimated according to a previous report [38].

## 3. Results

### 3.1. Difference in Callus Induction Rate and Callus-Based Root Regeneration among Peach Cultivars

To quantitatively assess the impact of genotype on callus induction, the size of callus induced from the designated cotyledon callus area together with the original cotyledon area were estimated (Figure 1A). The ratio of callus area to original cotyledon area was used to estimate CIR. A total of 100 peach cultivars were analyzed, and CIR showed a great variation among cultivars, ranging from 0–135% (Table S1). Based on their rates of callus induction, peach cultivars were divided into six classes: I (≥100%), II (70–99%), III (40–69%), IV (20–39%), V (1–19%) and VI (0%) (Figure 1B). Approximately 27% of cultivars showed no callus induction capacity, while one cultivar, BJLY, had the highest rate of callus induction. Both class II and class III cultivars exhibited relatively vigorous callus growth, accounting for 13% and 25%, respectively, of all accessions tested. By contrast, class IV and class V cultivars both had slow callus growth, and accounted for 16% and 17% of all tested accessions, respectively (Figure 1C).

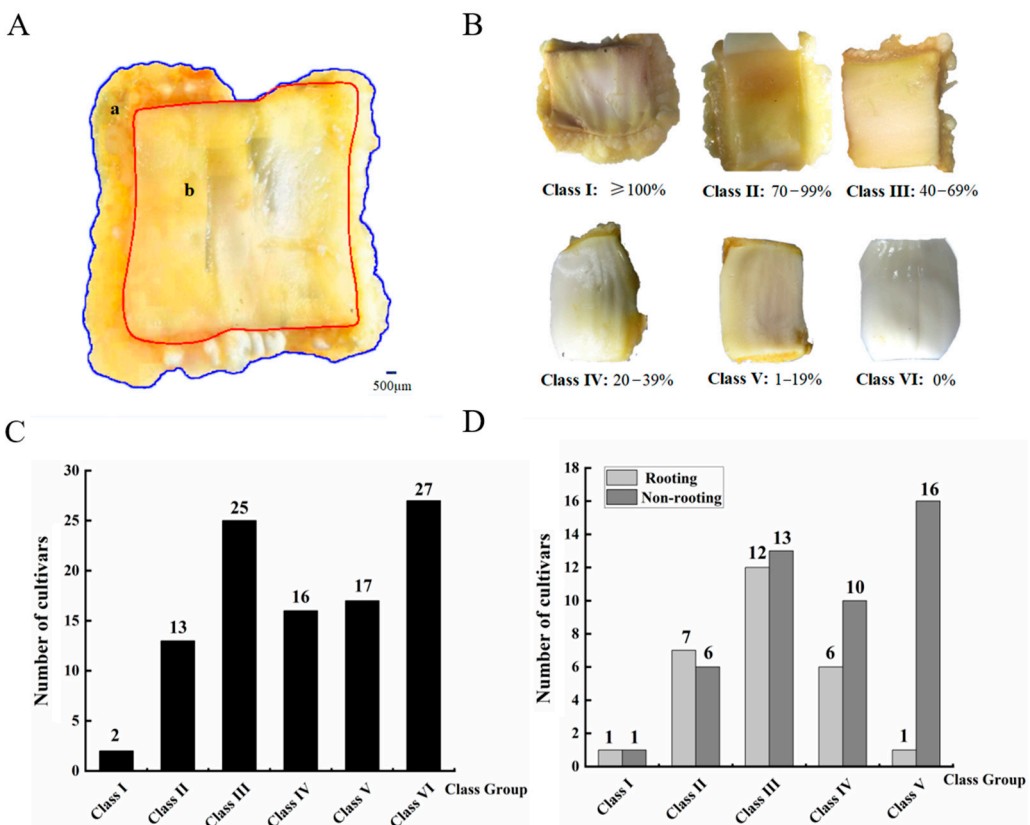

**Figure 1.** Relationship between callus induction rate and rooting ability. (**A**) Evaluation of rooting ability of callus from one hundred peach cultivars. Callus induction rate (CIR) was calculated based on the ratio of callus area (a) to original cotyledon area (b). (**B**) Classification of callus induction rate. (**C**) The numbers of cultivars in different CIR classes. (**D**) The numbers of cultivars exhibiting root regeneration ability in different CIR classes.

Notably, one or more ARs were often developed from the cotyledon-derived callus. An example of ARs developed from the callus of four cultivars is shown in Figure 2A. Approximately 50%, 54%, 48%, 38%, 6% of class I, class II, class III, class IV and class V cultivars, respectively, were capable of developing adventitious roots from cotyledon-derived callus, indicating an overall positive correlation between CIR and root regeneration capacity (Figure 1D). In addition, approximately 60% of the cotyledons from the ZYT cultivar developed adventitious roots during the process of callus induction. Interestingly, adventitious roots from the cotyledon-derived ZYT callus had a more vigorous growth

compared with those from the ZYT seedlings (Figure 2A,B), suggesting that ZYT has a strong propensity for adventitious rooting from callus.

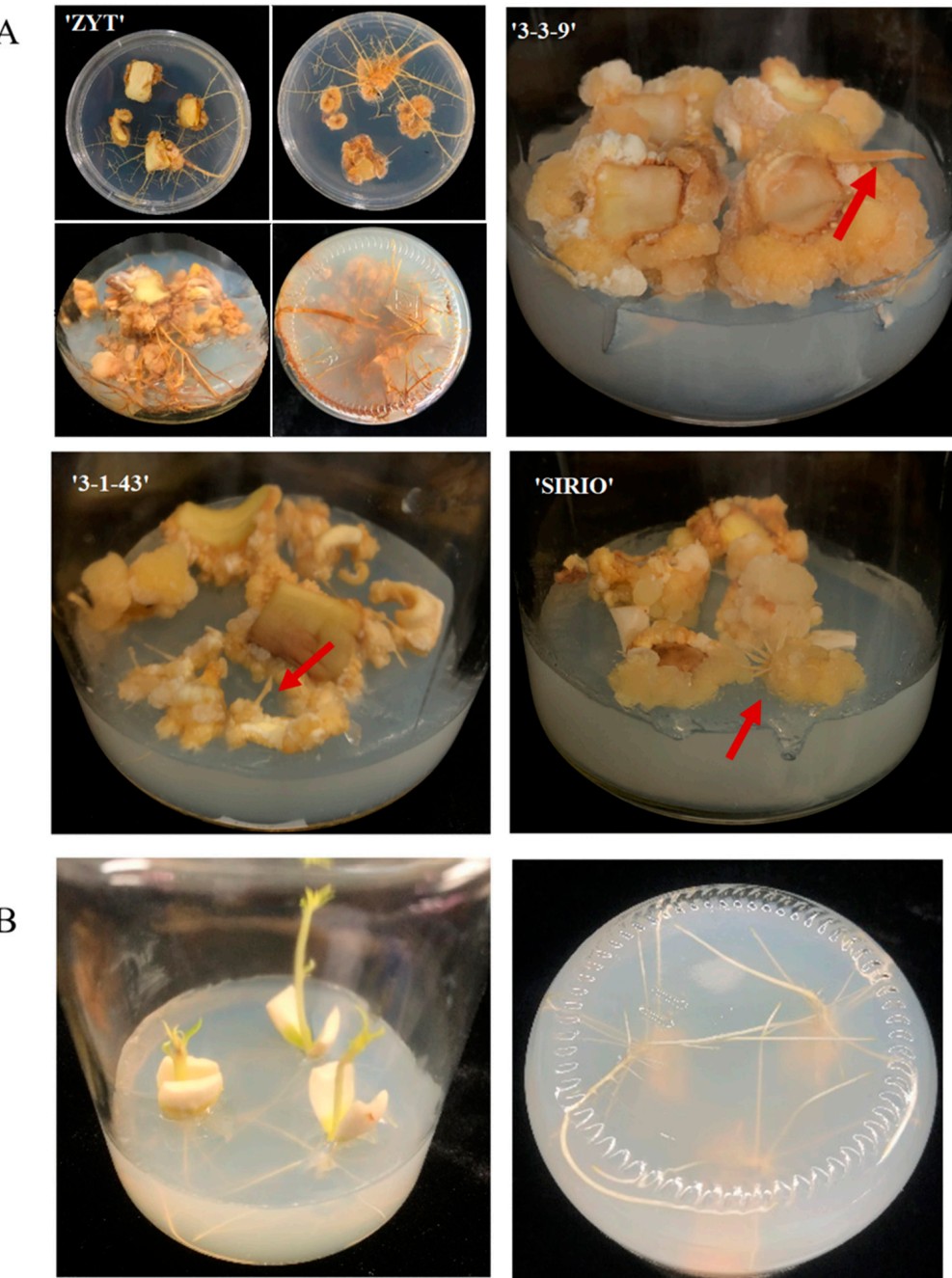

**Figure 2.** Adventitious root formation from cotyledon-derived callus from different peach cultivars. (**A**) Adventitious roots from cotyledon-derived callus from different peach cultivars cultured in CIM for 6 weeks. Note: the arrow shows that adventitious roots. (**B**) ZYT seedlings after cultivated on CIM for 6 weeks. The plastic petri dish is 90 mm in diameter, while the cultivation vial is 75 mm in diameter and 108 mm in height.

*3.2. Identification of Candidate Genes Related to Adventitious Root Formation during Callus Culture*

The CIR analysis revealed that cultivars ZH and QQ, like cultivar ZYT, were prone to developing adventitious roots from cotyledon-derived callus. Since long-term subcultured callus is characterized by a lack of regeneration ability [39], the five-year-old subcultured ZYT callus designated CC5Y-ZY was unable to regenerate adventitious roots. To investigate

the mechanism underlying callus-based root regeneration in peach trees, we compared the difference in transcriptome profile between the cotyledon-derived calli of ZYT, ZH, and QQ after 3 and 6 weeks of culture and CC5Y-ZY. For ease of description, cotyledon-derived calli with potential rooting capacity from ZYT, ZH and QQ after 3 weeks of culture in CIM were designated CC3W-ZY, CC3W-ZH, and CC3W-QQ, respectively, while the root-bearing calli from ZYT, ZH, and QQ after 6 weeks of culture on CIM were termed CC6WR-ZY, CC6WR-ZH, and CC6WR-QQ, respectively. The transcriptomes of CC3W-ZY/ZH/QQ, CC6WR-ZY/ZH/QQ, and CC5Y-ZY were sequenced, with each containing three biological replicates. After the adapter sequence and low-quality reads were removed, 143.33 Gb clean data were acquired from 21 RNA-seq libraries (Table S2). On average, approximately 93.8% of the clean reads were uniquely mapped to the reference genome Lovell v.2.0 [2] (Table S3).

A comparison of the transcriptomes from CC3W-ZY/ZH/QQ to those from CC5Y revealed 1773 and 1512 commonly up-regulated and down-regulated differentially expressed genes (DEGs), respectively (Figure 3A). Likewise, comparison of the transcriptomes from CC6WR-ZY/ZH/QQ with those from CC5Y revealed 2766 and 1605 commonly up-regulated and down-regulated DEGs, respectively (Figure 3A). Overall, 1621 commonly up-regulated DEGs and 1120 commonly down-regulated DEGs were expressed differentially in the pluripotent calli of the three cultivars tested and the long-term subcultured callus in the absence of totipotency (Table S4). Thus, these 2741 DEGs were expected to contain genes related to the formation of callus and adventitious rooting. Annotation of the 2741 DEGs was performed using the online program plantTFDB available at: http://planttfdb.gao-lab.org/ (accessed on 7 April 2022). As a result, 184 and 62 transcription factors (TFs) were identified in the commonly up-regulated and down-regulated DEGs, respectively (Table S5).

Of the 184 up-regulated TFs, 14 are homologous to previously reported positive regulators of root development in *Arabidopsis*, including *PpMYB6* (Prupe.7G216000), *PpRAX2* (Prupe.7G128800), *PpMYB114* (Prupe.3G042100), *PpANL2* (Prupe.2G291400), *PpARR2* (Prupe.6G254900), *PpLRP1* (Prupe.2G036500), *PpSRS3* (Prupe.8G088300), *PpbHLH123 (Prupe.4G196600), PpbHLH68 (Prupe.7G225800), PpDof1.4* (Prupe.4G042500), *PpDof2.4* (Prupe.6G079500), *PpERF109* (Prupe.8G125100), *PpLBD1* (Prupe.2G191100), and *PpWRKY75* (Prupe.1G223200) (Figure 3B; Table S6). The three *MYB* genes, *PpMYB6, PpRAX2,* and *PpMYB114*, are homologous to *AtMYB68,* involved in root development [40]; *AtMYB36,* controlling LR primordium (LRP) development [41]; and *AtMYB66,* related to the formation of root hair cell [42], respectively. Two *bHLH* genes, *PpbHLH123* and *PpbHLH68*, are homologs of *AtPFA3* and *AtPFA6*, which both determine lateral root initiation [43]. Two *Dof* TFs, *PpDof1.4* and *PpDof2.4*, are homologs of *AtURP3* and *AtDof2.4*, which are associated with the promotion of radial growth in the root apical meristem [44]. Two RING-like zinc finger TFs, *PpLRP1* and *PpSRS3*, are homologs of *AtLRP1* and *AtSTY1*, which control LRP development [45]. *PpANL2* is a homolog of the *Arabidopsis* HD-ZIP homeodomain protein gene *AtANL2*, which is concerned with root growth [46]. *PpARR2* is a homolog of the G2-like family gene *AtARR2*, which is related to the domination of root meristem growth [47]. An ethylene response factor (ERF) subfamily, B-3 of ERF/AP2 TF *PpERF109*, is homologous to *AtERF109*, which regulates lateral root formation by mediating cross-talk between jasmonic acid and auxin signaling [48], while *PpLBD1* is homologous to *AtLBD1*, which controls lateral organ fate [49]. The remaining gene, *PpWRKY75*, is a homolog of *AtWRKY75*, which is related to lateral root (LR) development [50].

Of the 62 down-regulated TFs, 6 are homologous to previously reported negative regulators of root development in *Arabidopsis*, including *PpPTI6* (Prupe.2G183200), *PpAIL6* (Prupe.6G203000), *PpARF17* (Prupe.1G507000), *PpREM1* (Prupe.8G122700), PpHAT4 (Prupe.5G064300), and *PpLBD4* (Prupe.4G138200) (Figure 3B; Table S6). *PpPTI6* is homologous to *Arabidopsis* CRF TFs, which modulate root and shoot growth [51] *PpAIL6* is a homolog of *AtPLT3*, the key regulator of the de novo organ model during *Arabidopsis* LR formation [52]. *PpARF17* is homologous to the auxin-inducible repressor *AtARF17*, which negatively regulates AR formation [53]. *PpREM1* is homologous to *AtREM1,* which is

involved in root gravitropic growth [54], while *PpHAT4* is homologous to *AtHAT4,* which inhibits root growth and branching [55]. *PpLBD4* is homologous to *AtLBD4*, a master regulator of root growth [56]. Taken together, above results suggested that 20 TFs have a potential function in the induction of callus-based adventitious roots in peach trees.

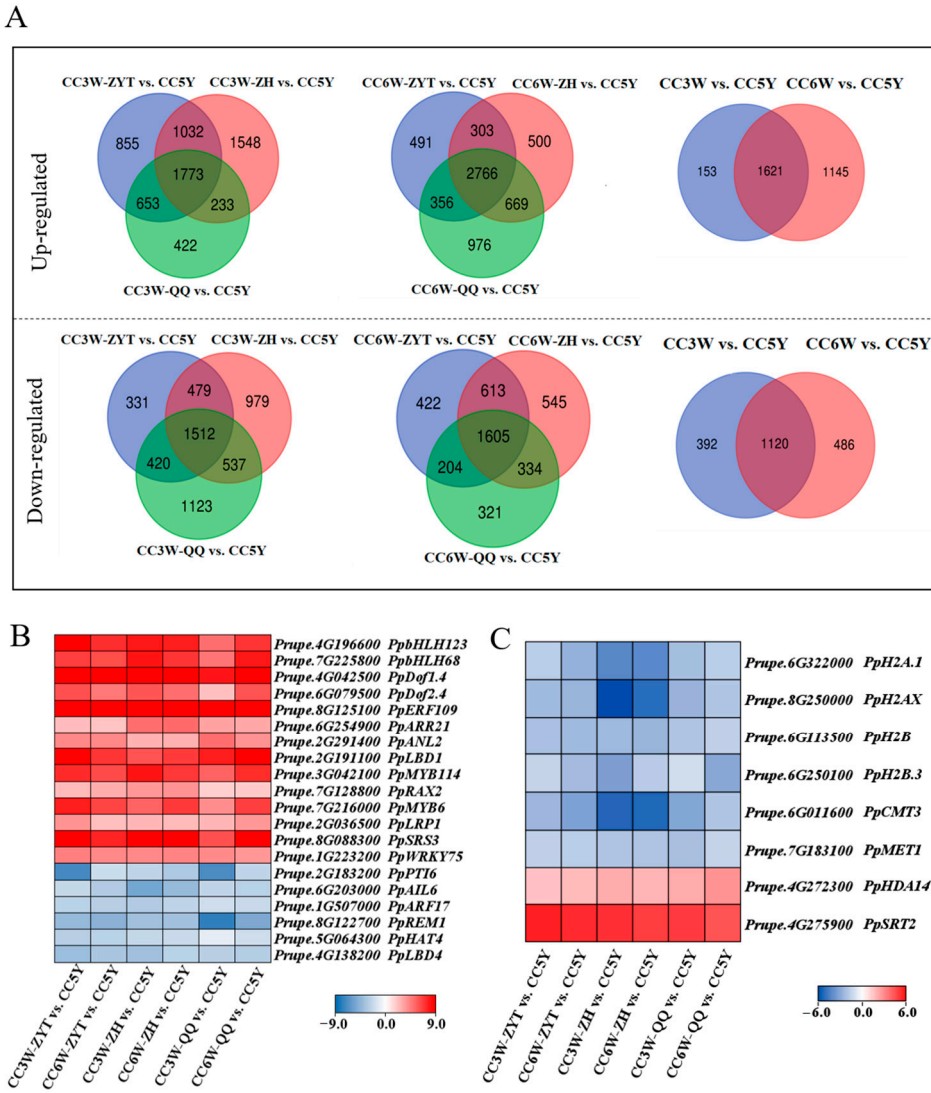

**Figure 3.** Identification of candidate genes related to the formation of callus adventitious rooting. (**A**) Venn diagram of DEGs between newly formed calli and long-term sub-cultured callus; (**B**) heat map display of transcription factors that were differentially expressed between newly formed calli and long-term sub-cultured callus; and (**C**) heat map display of DNA methylation and histone modification-related genes that were expressed differentially in newly formed calli and long-term sub-cultured callus.

Moreover, the abovementioned commonly up-regulated DEGs contained two histone deacetylase genes, *PpHDA14* and *PpSRT2* (Figure 3C) [57]. Similarly, the commonly down-regulated DEGs consisted of two DNA methyltransferase genes, *PpCMT3* and *PpMET1* [58], as well as four histone/histone variant genes, *PpH2A.1*, *PpH2AX*, *PpH2B*, and *PpH2B.3* [59] (Figure 3C). These results suggested a potential role in DNA methylation and histone modification in AR formation during peach callus culture.

### 3.3. Expression of Candidate Genes Associated with Callus Adventitious Rooting in the Cotyledon-Derived Callus

To confirm the relationship between callus adventitious rooting and the 20 abovementioned TFs, we investigated their expression in the cotyledon-derived callus of 8 cultivars, including 2, 2, and 4 of class I, class II, and class V cultivars, respectively. Of the 20 TFs, 6 showed high levels of expression in class I and class II cultivars, but were weakly expressed in class V cultivars (Figure 4), indicating a consistency between their expression and the callus rooting capacity. However, the remaining 14 TFs had no consistency between expression and callus rooting capacity (Figure S1). These results indicate that the six TFs *PpDof1.4*, *PpDof2.4*, *PpLBD1*, *PpSRS3*, *PpRAX2*, and *PpbHLH68* are good candidates for regulators correlated with the formation of ARs during callus culture in peach trees.

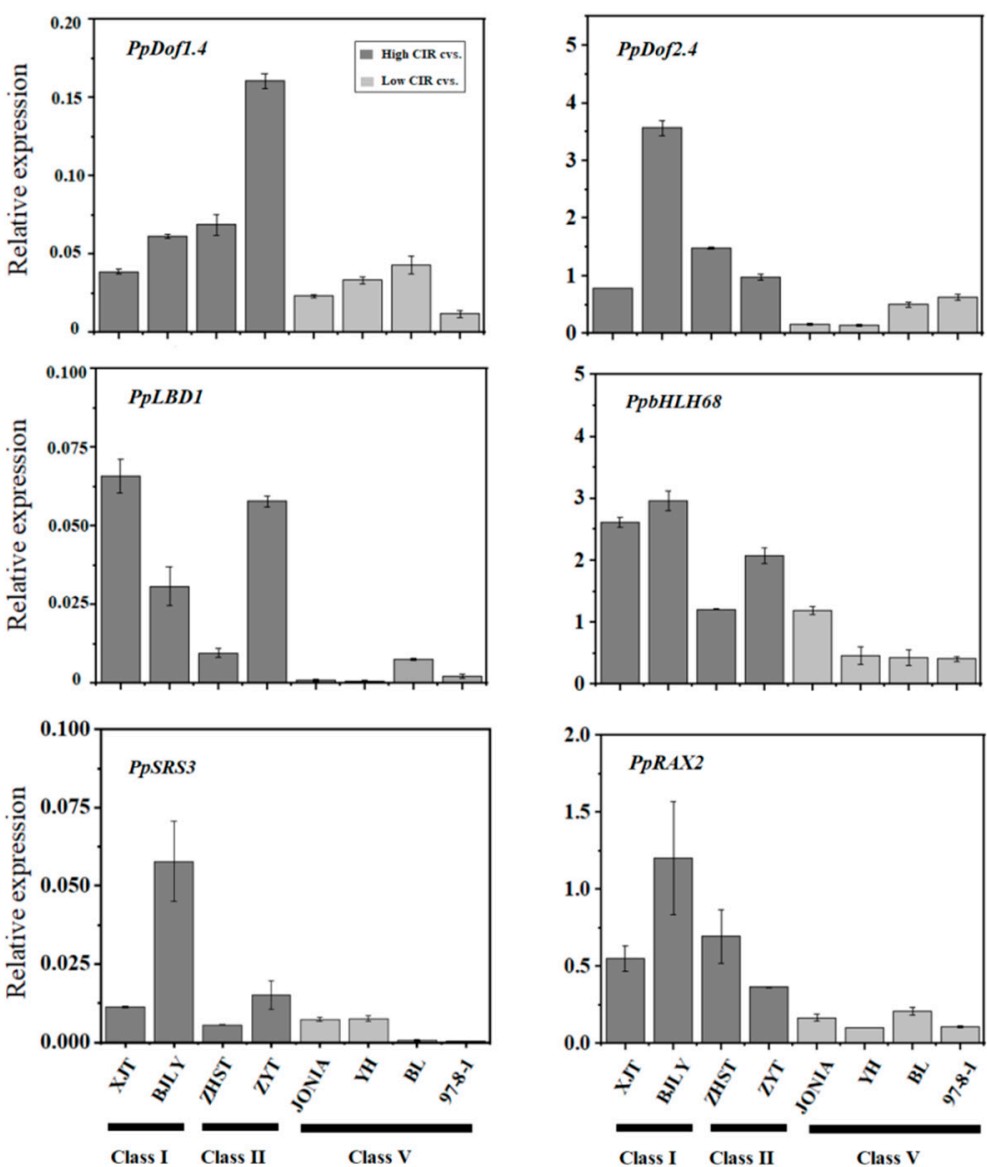

**Figure 4.** Expression levels of candidate genes associated with callus adventitious rooting in the cotyledon-derived callus. The genes expression as determined by Qrt-PCR.

### 3.4. Cellular Morphological Changes in the Cotyledon-Derived Callus during the Process of Root Differentiation

The abovementioned common DEGs consisted of genes relevant to sugar metabolism and flavonoid biosynthesis (Figure S2). Thus, we investigated metabolic changes in the cotyledon-derived callus during the process of root differentiation using high-performance liquid chromatography (HPLC) analysis. As a result, the soluble sugar contents of CC3W-ZY, CC6WR-ZY, and CC5Y-ZY all displayed the following trend: fructose > glucose > sucrose (Figure 5A). The CC5Y-ZY callus had the highest levels of fructose, glucose, and sucrose, corresponding to 7.91, 8.46 and 14.58 mg/g, respectively. By contrast, the contents of total phenols (TP) and flavonoids (FV) in CC5Y-ZY were extremely low, with 0.159 and 0.65 mg/g, respectively (Figure 5B). CC3W-ZY had the highest levels of TP and FV, while moderate levels of both TP and FV were observed in CC6WR-ZY.

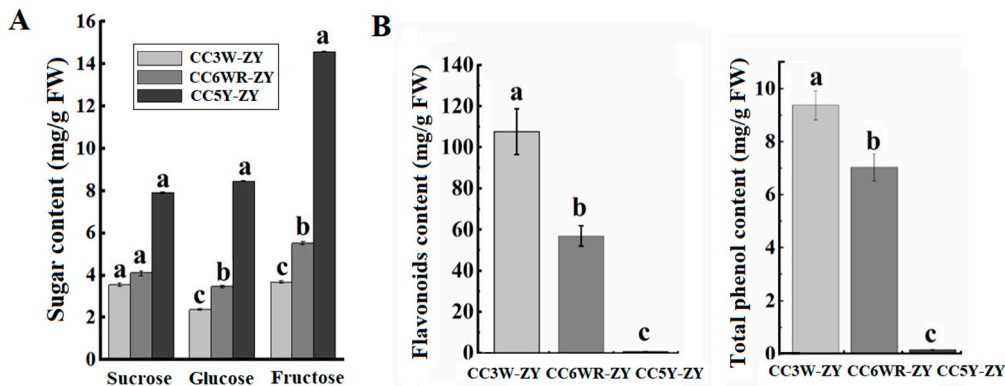

**Figure 5.** Metabolite content. (**A**) Soluble sugar content. (**B**) Total flavonoids and phenol content. The error bars show $\pm$ SE of at least three biological replicates, and significant difference at $p < 0.05$ is indicated by different lowercase letters based on LSD test.

Since the degradation of amyloplasts not only has an impact on sugar content, but is also associated with regeneration capacity [60], we investigated the cellular morphology of the cotyledon-derived callus at different stages. CC3W-ZY and CC6WR-ZY were yellow- and milky white-colored, respectively, while CC5Y-ZY had a glittering translucent appearance and was fragile (Figure 6A). Transmission electron microscopy results showed that CC3W-ZY cells had the smallest size and the largest number of amyloplasts, with abundant substances in the intercellular space (Figure 6B). CC6WR-ZY cells had a moderate size, fewer amyloplasts, and abundant phenolic compounds, with a moderate number of substances in the intercellular space. CC5Y-ZY cells were seriously vacuolated and contained many electron-dense inclusions, with no substances in the intercellular space. Notably, CC5Y-ZY cells had the largest number of mitochondria and an appearance of rough endoplasmic reticulum, suggesting an activation of cell metabolism and division. However, CC5Y-ZY cells lacked amyloplasts and obvious nuclei and nucleoli. These results suggested that the change in soluble sugars in the cotyledon-derived callus during the culture process is likely associated with the degradation of amyloplasts.

In addition, SEM analysis showed that CC3W-ZY was composed of plump round cells, while CC6WR-ZY cells had a columnar shape and loose structure (Figure 6C). CC5Y-ZY cells were irregular, elongated, and spiral-shaped. Both CC3W-ZY and CC6WR-ZY cells were tightly linked to each other with small intercellular spaces, but the reverse trend was observed in CC5Y-ZY cells. Altogether, these results suggest that the callus had undergone a significant change in cellular morphology during the culture process, and irregular morphology may be partially responsible for loss of regeneration capacity.

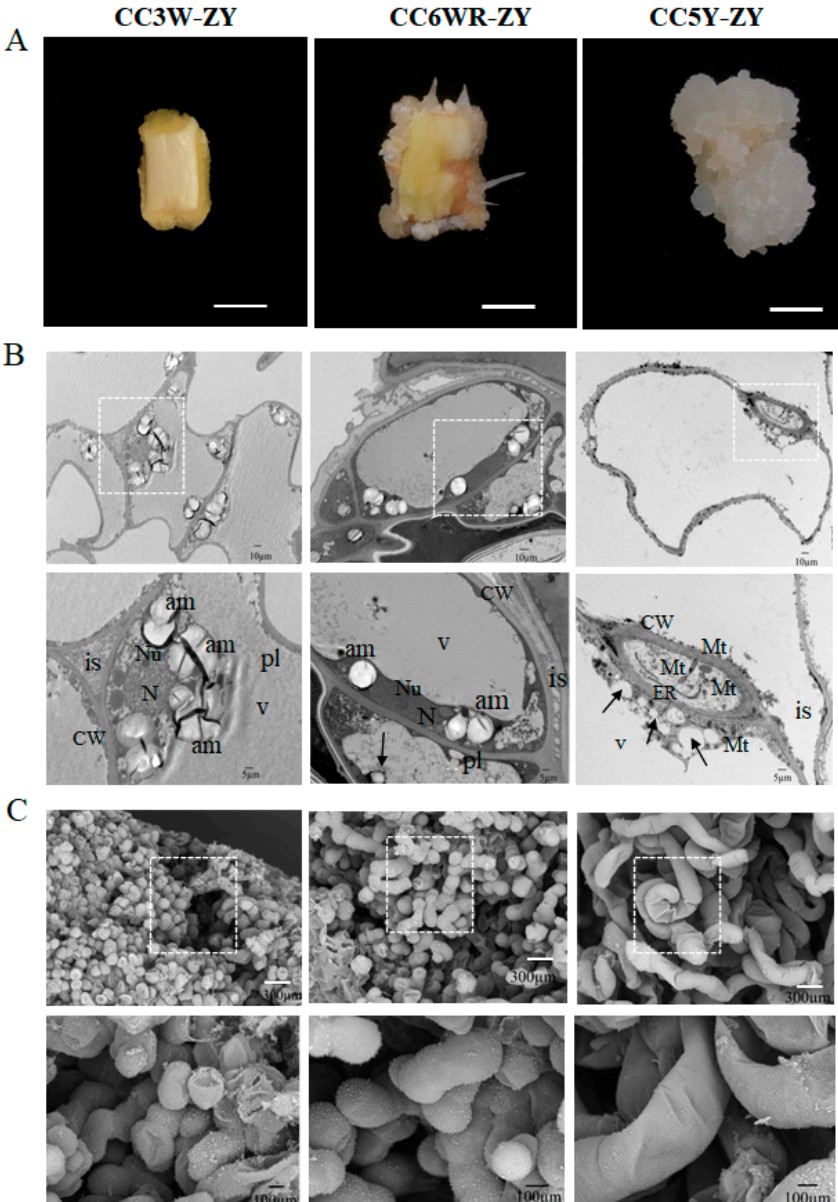

**Figure 6.** Morphology of CC3W-ZY, CC6WR-ZY, and CC5Y-ZY. (**A**) Appearance of ZYT cotyledon-derived callus culture in CIM at 3 weeks (CC3W-ZY), 6 weeks (CC6WR-ZY), and 5 years (CC5Y-ZY), respectively. (Bar = 1cm). (**B**) TEM images of three types of callus; magnification of the rectangle from subfigure (**B**) shows that cell ultra-structures are different. (**C**) SEM images of three callus types. Magnification of the rectangle from subfigure (**C**) shows that cell surfaces are different. Abbreviations: cell wall (CW), endoplasmic reticulum (ER), nucleus (N), nucleolus (Nu), mitochondria (Mt), phenolics (pl), vacuoles (v), amyloplasts (am), electron-dense inclusions (black arrows), intercellular space (is).

## 4. Discussion

### 4.1. Induction of Callus from Cotyledon and Its Root Regeneration in Peach trees

Callus can be rapidly propagated without the limitations of time and space and is thus considered one of the most important materials in verifying gene function [61]. Since callus induction is regarded as the critical stage for plantlet regeneration, callus induction rate is regarded as a critical index of regeneration ability [62]. As mentioned, callus has been successfully induced from leaf and stem explants in peach trees [30–32]; however, little is known about the influence of genotype on callus induction. In this project, we report the

first large-scale investigation of CIR in peach trees. Our results showed that callus induction rate is genotype-dependent in peach trees, which is consistent with previous reports in cotton [63], sugarcane [64], sunflowers [65], and roses [66]. Notably, late-maturing cultivars had a higher callus induction rate (>70%), while the majority of early maturing cultivars showed no ability to induce callus from cotyledon explants. Callus tested in this study was induced via cotyledon, which is a vital part of the seed. The physiological state of seeds affects the growth of cotyledons and ultimately the formation of callus. The seeds of early maturing cultivars were shrunken, while the seeds of late-maturing cultivars were plump. The cotyledons of the shrunken seeds were thin, semi-transparent, and irregular, which are the manifestations of incomplete development, while the plump seeds had fleshy and milky white-colored cotyledons with an oval shape. Thus, the low callus induction rate of early maturing cultivars is likely attributable to incomplete seed development. These results are consistent with a previous report that the physiological state of explants has an important function in callus induction [67].

Moreover, our results showed that genotypes with higher CIR had a tendency to develop adventitious roots during callus induction. Overall, callus induction rate was positively correlated with adventitious root regeneration ability among the tested peach cultivars. In *Arabidopsis*, the induction of gene correlation with the formation of lateral root primordium displays an important effect in callus induction [68,69]. Therefore, it seems that genes such as *PpLRP1* and *PpLBD16/29*, both involved in the formation of lateral root primordium, may have critical roles in callus induction in peach trees. It is worth noting that two peach cultivars, ZYT and XJT, had high callus induction rates and root regeneration ability. These two cultivars may be ideal genotypes for the construction of the peach regeneration system in the future.

### 4.2. PpLBD1 Is Likely Involved in Callus Adventitious Rooting in Peach Trees

Callus cells have high plasticity when differentiating into adventitious shoot and root via direct somatic embryogenesis (SE) [6]. Previous studies have indicated that callus is induced via the LR development route [68,70]. The newly formed callus cells resemble root primordium-like cells that have potential adventitious root regeneration ability. Here, the five-year-old subcultured callus lost the potential to regenerate roots. Transcriptome comparison of the five-year-old subcultured callus with the newly formed callus revealed six candidate genes involved in callus adventitious rooting. These include an LBD family TF *PpLBD1*, which exhibited a high expression in the newly formed callus but was undetected in the five-year-old subcultured callus. Validation in different cultivars showed that *PpLBD1* might positively regulate the rooting of peach callus. Conversely, in this study, *PpLBD16* and *PpLBD29* showed a high expression in five-year-old subcultured callus, but a weak expression in the newly formed callus. The *LBD1* gene has been proven to control the development and architecture of lateral roots in *Medicago truncatula* and to regulate of secondary growth in *Populus* [71,72]. In *Arabidopsis,* LBD16/29 can form a complex with AtbZIP59 to activate transcription of the *FAD-BD* gene, thereby promoting the formation of callus [73]. Thus, *PpLBD1* may be a good candidate gene bound up with the regulation of callus adventitious rooting in peach trees, while *PpLBD16* and *PpLBD29* may be at least partially in charge of the strong callus proliferation ability of the five-year-old subcultured callus.

### 4.3. The Effect of Metabolites on Callus Adventitious Rooting in Peach

Starch is the most universal carbon form for energy substrate, and starch granules in callus serve as an energy source for the regeneration procedure [74–76]. In this study, starch granules were abundant in the newly formed callus with root regeneration ability, while the five-year-old subcultured callus contained few starch granules. This suggests that the deposition of starch granules is crucial for root regeneration from callus. Granule-bound starch synthase 1 (GBSS1) catalyzes one of the enzymatic steps of starch synthesis. *PpGBSS1* (Prupe.5G132800) exhibited a highly expression in the newly formed callus, but

was weakly expressed in the five-year-old subcultured callus. Thus, activation of *GBSS1* is likely required for root regeneration from callus.

In addition to starch granules, flavonoid content showed a significant difference between the newly formed callus and the five-year-old subcultured callus. The regulatory roles of flavonoids in adventitious rooting of stem cuttings are different between plant species. For example, in cherry and chestnut, flavonoid accumulation inhibits AR regeneration in stem cuttings [77,78]. However, flavonoid biosynthesis has a positive effect on AR formation of stem cuttings in olive [79]. In *Chamaelaucium uncinatum*, flavonoids negatively and positively regulate the formation of ARs from hardwood and softwood cuttings, respectively [80]. In this study, the flavonoid content in the five-year-old subcultured callus was significantly lower than in the newly formed callus, suggesting that flavonoid accumulation might have a positive regulatory role in root regeneration from callus. Among the flavonoid biosynthesis genes, *PpPAL* (Prupe.6G235400) encoding phenylalanine ammonia lyase (PAL) displayed a diversity in expression between the newly formed callus and the five-year-old subcultured callus. This finding is in agreement with the previous report that PAL is the rate-limiting enzyme of flavonoid biosynthesis [81]. Hence, changes in *PpPAL* expression may cause differences in flavonoid levels and affect the formation of ARs in peach callus.

### 4.4. Epigenetic Regulation in Callus Adventitious Rooting in Peach

Cell morphology changes significantly during the process of subculture, caused by the loss of callus regeneration ability [82]. In *Theobroma cacao*, global DNA methylation levels show an increase in aged somatic embryos after long-term in vitro culture, but this increase can be inhibited via Aza treatment [83]. Thus, epigenetic regulation seems to show a crucial role in determining callus cell morphology. In this study, the cells of five-year-old callus exhibited irregular, elongated, and spiral-shaped morphology. Moreover, the expression of genes involved in DNA methylation (*PpCMT3* and *PpMET1*) and histone methylation (*PpH2A.1*, *PpH2AX*, *PpH2B*, and *PpH2B.3*) was found to be up-regulated in the five-year-old callus compared to the newly formed callus. This suggests that long-term subculture callus in peach trees has a significant change in epigenetic modifications, which coincides with a previous report that numerous of epigenetic modifications occur during the process of callus-based regeneration in plant [84]. Change in epigenetic landscape is known to be accompanied by transition of cell fate [85,86]. Here, our results indicate that aged callus cells with long-term in vitro culture lost root regeneration ability. Therefore, epigenetic regulation is likely associated with callus adventitious rooting capacity in peach trees.

### 5. Conclusions

In this study, a significant difference in CIR was found among peach germplasms. Notably, the cultivars ZYT and XJT can be recommended as ideal candidates for constructing regeneration system in peach. Moreover, our comparative transcriptome analysis revealed *PpLBD1* as a candidate gene implicated in the AR development. Overall, the results of this study not only provide valuable insights into the underlying mechanisms of regeneration, but are also useful for the development of regeneration systems in peach trees.

**Supplementary Materials:** The following supporting information can be downloaded at: https://www.mdpi.com/article/10.3390/horticulturae9080850/s1, Figure S1: Expression levels of candidate genes associated with callus adventitious rooting in the cotyledon-derived callus. The genes expression as determined by Qrt-PCR; Figure S2: Gene ontology (GO) enrichment; Table S1: Peach germplasm used in this study; Table S2: Transcriptome sequencing data in *Prunus persica* L.; Table S3: Transcriptome sequencing data in *Prunus persica* L.; Table S4 Total 2741 DEGs were divided into up- and down-regulated according to log2FC; Table S5 Transcription factors distribution in up- and down-regulated; Table S6 Candidate genes that may be related to adventitious roots formation of cotyledon-derived callus in peach; Table S7 Real-time quantitative PCR primers used in this study.

**Author Contributions:** L.G.: Conceptualization, Writing—original draft; Methodology and Validation. J.L.: Data analysis and project administration. L.L.: Data curation, Software and Visualization. A.G.: Validation and collect samples. B.N.N.: Validation and writing. C.Z.: Investigation, Provision and collection resources. B.Z.: Writing—review & editing, designed the experiments. Y.H.: Writing—review & editing and supervision. All authors have read and agreed to the published version of the manuscript.

**Funding:** This research was supported by funding received from National Natural Science Foundation of China (32272690 and 32272687), the National Key R&D Program of China (2019YFD1000800), and the China Agriculture Research System (grant no. CARS-30).

**Institutional Review Board Statement:** Not applicable.

**Informed Consent Statement:** Not applicable.

**Data Availability Statement:** Download the supplementary figure and table files used in this project online.

**Conflicts of Interest:** The authors declare no conflict of interest.

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
