# Peer review of "Callus Induction and Adventitious Root Regeneration of Cotyledon Explants in Peach Trees"

_horticulturae, doi:10.3390/horticulturae9080850_

Round 1
Reviewer 1 Report
Dear Authors,
I have reviewed the manuscript you have prepared and I think that this work will make useful contributions to the literature. However, after the following recommendations are met on the manuscript in the attached document, the manuscript can be publishable form in the journal.
# In the Introduction section, sentences containing some definite information require reference citiation. These are marked separately in the attached manuscript.
# In the method section, information on the surface sterilization method and medium composition, callus induction rate formula used in this study should be stated, whether it was optimized or developed in this study or used from a previous study. If taken from previous studies, it should be cited.These are indicated on the text in the attached document. I did not see the table provided in S7, so it should be noted whether the selected primers were designed from the NCBI (or another) database or taken from a previous study. Reference should be cited for each primer if taken from a previous study.
# Discussion and conclusions should not be given under the same heading. Conclusions should be given in a separate title. A great deal of data has been obtained from the study and the discussion section can be expanded further by comparing these data with current references.
# References should be arranged according to the writing rules of the journal. https://www.mdpi.com/journal/horticulturae/instructions
Kind regards

Manuscript requires very minor grammatical corrections.
Author Response
I have reviewed the manuscript you have prepared and I think that this work will make useful contributions to the literature. However, after the following recommendations are met on the manuscript in the attached document, the manuscript can be publishable form in the journal.
Response: We appreciate the reviewer’s comments.
# In the Introduction section, sentences containing some definite information require reference citation. These are marked separately in the attached manuscript.
Response: The introduction section has been modified with the addition of reference citations according to the reviewer's valuable suggestion. Please check lines 29, 37 and 47.
Moreover, comments and suggestions marked in the attached manuscript also has been revised. Please check lines 42, 50-53, 56, 58, 93 and 260.
# In the method section, information on the surface sterilization method and medium composition, callus induction rate formula used in this study should be stated, whether it was optimized or developed in this study or used from a previous study. If taken from previous studies, it should be cited. These are indicated on the text in the attached document. I did not see the table provided in S7, so it should be noted whether the selected primers were designed from the NCBI (or another) database or taken from a previous study. Reference should be cited for each primer if taken from a previous study.
Response: In the method section, the surface sterilization method was optimized from our previous study. Medium composition was reported in another our study. Callus induction rate formula was developed in this study. The related references were added according to the reviewer’s suggestion. Please check line 75 and 96.
Primer sequences were listed in Table S7. The information of the selected primers was noted according to the reviewer’s comments. Please check Table S7 in the supplementary table file.
# Discussion and conclusions should not be given under the same heading. Conclusions should be given in a separate title. A great deal of data has been obtained from the study and the discussion section can be expanded further by comparing these data with current references.
Response: Conclusions was given in a separate title according to the reviewer’s suggestion. Please check lines 409-414.
In the discussion section, we expanded further by comparing these data with current references according to the reviewer’s comments. Please check lines 391-407. Meanwhile, we also supplemented the corresponding results. Please check lines 245-250.
# References should be arranged according to the writing rules of the journal. https://www.mdpi.com/journal/horticulturae/instructions
Response: References were revised according to the writing rules of the journal. Thanks for the reviewer’s kind remind. Please check the references section in this manuscript.

Reviewer 2 Report
Dear Editor,
The manuscript entitled "Callus induction and adventitious root regeneration of cotyledon explants in peach" submitted to Horticulturae describes the diversity of callus induction rates in 100 peach cultivars and the identification of genes involved in adventitious root development. After careful review of the manuscript, I have observed that manuscript needs to be thoroughly revised before reconsideration for publication in Horticulturae. There are discrepancies, unnecessary repetitions and linguistic errors in many parts of the manuscript that should definitely be improved.
Some comments are listed below:
- English improvement is required. For example:
Please, rewrite the sentences in order to be written correctly in English (Lines 7-11, 72-77).
Line 28, “development” should be “develop”
Line 85, “cultured” should be “culture”
Line 112, “isolated” should be “isolation”
Lines 118-119, please rewrite this sentence “All the samples were bioreplicates for three times”
- Unnecessary repetitions. For example, lines 123-125, this sentence belongs in the Material and Methods section.
- Supplementary files are not available.
Abstract:
- Line 9 – “two high CIR varieties” but three are listed in brackets.
Material and methods:
- Please add a separate subsection on transcriptome analysis and gene identification. The authors did not describe RNA sequencing and transcriptome analysis in Material and methods section .
Results:
- Figure 1C: “number of cultivars” should be “callus induction rate”? Besides, Fig 1C is not mentioned in the text.
- Line 155, abbreviations without explanation, e.g. cvs and cv
- English improvement is required. For example:
Please, rewrite the sentences in order to be written correctly in English (Lines 7-11, 72-77).
Line 28, “development” should be “develop”
Line 85, “cultured” should be “culture”
Line 112, “isolated” should be “isolation”
Lines 118-119, please rewrite this sentence “All the samples were bioreplicates for three times”
Author Response
Comments and Suggestions for Authors
Dear Editor,
The manuscript entitled "Callus induction and adventitious root regeneration of cotyledon explants in peach" submitted to Horticulturae describes the diversity of callus induction rates in 100 peach cultivars and the identification of genes involved in adventitious root development. After careful review of the manuscript, I have observed that manuscript needs to be thoroughly revised before reconsideration for publication in Horticulturae. There are discrepancies, unnecessary repetitions and linguistic errors in many parts of the manuscript that should definitely be improved.
Response: We revised our manuscript according to the reviewer’s comments.
Some comments are listed below:
#English improvement is required. For example:
Please, rewrite the sentences in order to be written correctly in English (Lines 7-11, 72-77).
Response: The sentences were revised according to the reviewer’s suggestion. Please check lines 8-11 and 74-83.
Line 28, “development” should be “develop”
Response: ‘development’ was changed to ‘develop’. Please check line 30.
Line 85, “cultured” should be “culture”
Response: ‘cultured’ was changed to ‘culture’. Please check line 92.
Line 112, “isolated” should be “isolation”
Response: ‘isolated’ was changed to ‘isolation’. Please check line 118.
Lines 118-119, please rewrite this sentence “All the samples were bioreplicates for three times”
Response: The sentence was revised according to the reviewer’s suggestion. Please check lines 124-125.
Unnecessary repetitions. For example, lines 123-125, this sentence belongs in the Material and Methods section.
Response: The repetition was deleted according to the reviewer’s suggestion. Please check lines 139-142.
Supplementary files are not available.
Response: Supplementary files were provided and we are sorry for the mistake. Please check the supplementary files.
Abstract:
Line 9 – “two high CIR varieties” but three are listed in brackets.
Response: Abstract was revised and this mistake was revised.
Material and methods:
Please add a separate subsection on transcriptome analysis and gene identification. The authors did not describe RNA sequencing and transcriptome analysis in Material and methods section.
Response: The transcriptome analysis and gene identification subsection was added to M&M according to reviewer’s comments. Please check lines 126-135.
Results:
Figure 1C: “number of cultivars” should be “callus induction rate”? Besides, Fig 1C is not mentioned in the text.
Response: Figure 1C shows the number of cultivars in different classes of CIR, abscissa axis indicates callus induction rate which have been shown in Figure 1B, and vertical axis indicates numbers of cultivar in each class of CIR. Fig 1C has been mentioned in the text according to the reviewer’s comments. Please check line 150.
Line 155, abbreviations without explanation, e. g. cvs and cv
Response: Abbreviations of cvs. and cv. represent cultivar and cultivars, respectively. These abbreviations were revised according to the reviewer’s comments. Please check lines 175.

Reviewer 3 Report
The manuscript focuses on adventitious regeneration in peach Prunus persica L., comparing the regeneration in older and new tissue, based on transcriptomic and cytological (microscopy) analysis. Remarkable amount of cultivars were analysed. Although it contains interesting information (especially cytology part), some parts need to be improved (especially M&M section and almost all Figures). English also needs to be polished in some parts of MS. I have some more additional comments and questions.
Here are my comments:
Line 39 – rephrase as “as shown in our study”
line 63 – rephrase as “also contribute”
line 70-77 – please keep the passive tense through whole section
line 79 – rephrase whole sentence in a way to avoid parentheses
line 80 – maybe I didn´t see it, but what type of medium was it? Add type and citation and pH of medium ready-to-use. Also, add conditions you have in autoclave (xx minutes, xx degrees Celsius etc.). Why did you choose this specific PGRs? Define, and cite if needed.
line 88 – “Fig 1A” does not belong in M&M section. Please delete.
line 93 – SEM – abbreviation is not defined
lines 111-113 – Chinese towns should start with capitals
line 118 – add citation for this method, if needed (cycle threshold (Ct) 2–ΔΔCt method)
line 136 – “100 Peach Germplasm“ – correct as „one hundred peach germplasm“
line 139 – here you have variety, in the graph it is cultivar – unify in the whole manuscript
Figure 1 – I have some issues with this picture. First, picture A – can you either add scale or length/width of explant? I know it´s in M&M, but I think it is useful also here. Second, I would not repeat the formula again, it´s just a few lines above the Figure and also should be in M&M section as all other formulas. Third, “cotyledon-derived callus of 27 peach cultivars can form roots“ needs to be rephrased. It should be description, not result.
Figure 2 – scale is missing, or at least add the diameter of cultivation vessel in the caption. Also, caption is very short. Add info such as – the length of cultivation, PGR used in medium, what medium, etc.
line 155 – in one part of manuscript you have cv. ZYT, in another Ê»ZYTʼ. Please unify in the whole manuscript.
line 185 – Venn diagram, not Veen
line 242 – I believe that if you used HPLC, it should be mention also in M&M
Figure 5 – here you have “Fig.5”, while in the rest is “Figure” – please unify in the whole manuscript. On picture A, a b c.. is not bold and seems to be in different font
line 250 – soluble sugar
line 267 – move this definition to line 93 (first mention of SEM)
line 278 and 279 – callus types
line 282 – amyloplasts
line 283 – I am sorry but I don´t see Conclusion here. If you don´t put separate part “Conclusion”, or you don´t add Conclusion as the last section in your Discussion, the title should be just Discussion. It´s up to your decision, unless is Conclusion obligatory (see the guidelines)
line 309 – rephrase as “Therefore, it seems that genes such as PpLRP1 and PpLBD16/29, both involved in the formation of lateral root primordial, may have critical roles in callus induction in peach.“
line 322 – rephrase and correct these part “Among these candidate genes, including an LBD family TF PpLBD1. PpLBD1 showed a high expression in the newly-formed callus, but not detected in the five-year-old subcultured callus.“
Comments to Supplementary Files:
Table S1 – please define what is „ / “ in the Table.
Table S4 – Caption – please add „that were divided...“
Table S5 – the Caption seems to be missing something, please correct
Table S6 – Caption – please add „that may be related..“
it is all in the document.
Author Response
Comments and Suggestions for Authors
The manuscript focuses on adventitious regeneration in peach Prunus persica L., comparing the regeneration in older and new tissue, based on transcriptomic and cytological (microscopy) analysis. Remarkable amount of cultivars were analysed. Although it contains interesting information (especially cytology part), some parts need to be improved (especially M&M section and almost all Figures). English also needs to be polished in some parts of MS. I have some more additional comments and questions.
Response: We revised the manuscript according to the reviewer’s comments.
Here are my comments:
Line 39 – rephrase as “as shown in our study”
Response: The sentence was revised according to the reviewer’s comments. Please check lines 42.
line 63 – rephrase as “also contribute”
Response: ‘also contributes’ was changed to ‘also contribute’ according to the reviewer’s comments. Please check line 67.
line 70-77 – please keep the passive tense through whole section
Response: The passive tense was used through whole section according to the reviewer’s comments. Please check lines 75-83.
line 79 – rephrase whole sentence in a way to avoid parentheses
Response: The MS medium was the same as our previous report and the related reference was cited. Please check lines 85-86.
line 80 – maybe I didn´t see it, but what type of medium was it? Add type and citation and pH of medium ready-to-use. Also, add conditions you have in autoclave (xx minutes, xx degrees Celsius etc.). Why did you choose this specific PGRs? Define, and cite if needed.
Response: The composition of the medium used in this study was adopted from our previous research (Zheng et al., 2022. Horticulture Research). This particular medium, supplemented with the specified plant growth regulators (PGRs), has demonstrated successful results in callus induction. In accordance with the reviewer's suggestion, we have included reference citations for the previous study. Additionally, the autoclave conditions were adjusted to sterilize the medium by autoclaving at 121ºC for 20 minutes, as the reviewer's recommendation. Please check lines 85-88.
line 88 – “Fig 1A” does not belong in M&M section. Please delete.
Response: The repetition was deleted according to the reviewer’s suggestion. Please check line 95.
line 93 – SEM – abbreviation is not defined
Response: Scanning electron microscope was abbreviated as ‘SEM’, we have defined this abbreviation according to the reviewer’s suggestion. Please check lines 100.
lines 111-113 – Chinese towns should start with capitals
Response: We revised Chinese towns name first letter to capitals according to the reviewer’s suggestion. Please check lines 117-119.
line 118 – add citation for this method, if needed (cycle threshold (Ct) 2–ΔΔCt method)
Response: We added cycle threshold (Ct) 2–ΔΔCt method citation reference (Livak et al., 2001. Methods) according to the reviewer’s suggestion. Please check lines 129.
line 136 – “100 Peach Germplasm” – correct as “one hundred peach germplasm”
Response: We corrected ‘100 Peach Germplasm’ as ‘one hundred peach germplasm’ according to the reviewer’s suggestion. Please check lines 153.
line 139 – here you have variety, in the graph it is cultivar – unify in the whole manuscript
Response: We unified expression of cultivar in the whole manuscript according to the reviewer’s suggestion. Please check lines 155 and 168.
Figure 1 – I have some issues with this picture. First, picture A – can you either add scale or length/width of explant? I know it´s in M&M, but I think it is useful also here. Second, I would not repeat the formula again, it´s just a few lines above the Figure and also should be in M&M section as all other formulas. Third, “cotyledon-derived callus of 27 peach cultivars can form roots” needs to be rephrased. It should be description, not result.
Response: For picture A, we added the scale of explant according to the reviewer’s suggestion. Please check lines 151. The formula in Figure 1A caption had been deleted according to the reviewer’s suggestion. Please check lines 153-154. Third, we rephrased the description of Figure 1D according to the reviewer’s suggestion. Please check lines 155-156.
Figure 2 – scale is missing, or at least add the diameter of cultivation vessel in the caption. Also, caption is very short. Add info such as – the length of cultivation, PGR used in medium, what medium, etc.
Response: The size information of cultivation vessel and plastic petri dishes were added in the caption. Meanwhile, modified the caption with cultivation time and medium according to the reviewer’s suggestion. Please check lines 170-171.
line 155 – in one part of manuscript you have cv. ZYT, in another Ê»ZYTʼ. Please unify in the whole manuscript.
Response: We unified the ‘ZYT’ name in the whole manuscript according to the reviewer’s suggestion. Please check lines 71, 175.
line 185 – Venn diagram, not Veen
Response: ‘Veen’ was changed to ‘Venn’ according to the reviewer’s comments. Please check line 205.
line 242 – I believe that if you used HPLC, it should be mention also in M&M
Response: We used HPLC method as our previous reported to measure soluble sugars in our previous study (Zhen et al., 2018. Horticulture Research). And the citation reference was list in the manuscript. M&M was revised according to the reviewer’s suggestion. Please check line 133-134 and lines 535-537.
Figure 5 – here you have “Fig.5”, while in the rest is “Figure” – please unify in the whole manuscript. On picture A, a b c. is not bold and seems to be in different font
Response: We unified the ‘Figure’ name in the whole manuscript. Please check lines 152, 168, 204, 263, 280 and 305. The font of picture A was revised according to the reviewer’s suggestion. Please check line 279.
line 250 – soluble sugar
Response: ‘Soluble Sugar’ was changed to ‘soluble sugar’ according to the reviewer’s comments. Please check line 280.
line 267 – move this definition to line 93 (first mention of SEM)
Response: We moved this definition to the first mention of SEM according to the reviewer’s suggestion. Please check line 100.
line 278 and 279 – callus types
Response: ‘three types callus’ was changed to ‘three callus types’ according to the reviewer’s comments. Please check line 308-309.
line 282 – amyloplasts
Response: The word was revised according to the reviewer’s suggestion. Please check lines 311.
line 283 – I am sorry but I don´t see Conclusion here. If you don´t put separate part “Conclusion”, or you don´t add Conclusion as the last section in your Discussion, the title should be just Discussion. It´s up to your decision, unless is Conclusion obligatory (see the guidelines)
Response: We added conclusion section and changed the discussion section name according to the reviewer’s comments. Please check lines 313 and 408-414.
line 309 – rephrase as “Therefore, it seems that genes such as PpLRP1 and PpLBD16/29, both involved in the formation of lateral root primordial, may have critical roles in callus induction in peach”.
Response: The sentences were rephrased according to the reviewer’s comments. Please check lines 339-341.
line 322 – rephrase and correct these part “Among these candidate genes, including an LBD family TF PpLBD1. PpLBD1 showed a high expression in the newly-formed callus, but not detected in the five-year-old subcultured callus.“
Response: The sentences were rephrased and corrected according to the reviewer’s comments. Please check lines 352-354.
Comments to Supplementary Files:
Table S1 – please define what is „ / “ in the Table.
Response: The definition of ‘ / ‘ means these cultivars not yet induced callus, we added the definition according to the reviewer’s comments. Please check in Table S1.
Table S4 – Caption – please add „that were divided...“
Response: The caption was added according to the reviewer’s comments. Please check in Table S4.
Table S5 – the Caption seems to be missing something, please correct
Response: The caption was added according to the reviewer’s comments. Please check in Table S5.
Table S6 – Caption – please add „that may be related..“
Response: The caption was added according to the reviewer’s comments. Please check in Table S6.

Round 2
Reviewer 3 Report
Dear authors, thank you for the revision. The only issue I found is that in Abstract and Figure Captions, the added text is bigger than the rest of text in MS. Please correct. That is all, good luck